# Positive Externalities of Climate Change Mitigation and Adaptation for Human Health: A Review and Conceptual Framework for Public Health Research

**DOI:** 10.3390/ijerph18052481

**Published:** 2021-03-03

**Authors:** Jean C. Bikomeye, Caitlin S. Rublee, Kirsten M. M. Beyer

**Affiliations:** 1PhD Program in Public and Community Health, Institute for Health and Equity, Medical College of Wisconsin, 8701 Watertown Plank Rd., Milwaukee, WI 53226, USA; jbikomeye@mcw.edu; 2Department of Emergency Medicine, Medical College of Wisconsin, 8701 Watertown Plank Rd., Milwaukee, WI 53226, USA; crublee@mcw.edu; 3Division of Epidemiology, Institute for Health and Equity, Medical College of Wisconsin, 8701 Watertown Plank Rd., Milwaukee, WI 53226, USA

**Keywords:** climate change, mitigation and adaptation, greenhouse gas emissions (GHGE), health co-benefits, global health, public health, green infrastructure, conceptual framework, health equity, positive externalities, Paris agreement, nationally determined contributions (NDC), extreme weather events (EWE)

## Abstract

Anthropogenic climate change is adversely impacting people and contributing to suffering and increased costs from climate-related diseases and injuries. In responding to this urgent and growing public health crisis, mitigation strategies are in place to reduce future greenhouse gas emissions (GHGE) while adaptation strategies exist to reduce and/or alleviate the adverse effects of climate change by increasing systems’ resilience to future impacts. While these strategies have numerous positive benefits on climate change itself, they also often have other positive externalities or health co-benefits. This knowledge can be harnessed to promote and improve global public health, particularly for the most vulnerable populations. Previous conceptual models in mitigation and adaptation studies such as the shared socioeconomic pathways (SSPs) considered health in the thinking, but health outcomes were not their primary intention. Additionally, existing guidance documents such as the World Health Organization (WHO) Guidance for Climate Resilient and Environmentally Sustainable Health Care Facilities is designed primarily for public health professionals or healthcare managers in hospital settings with a primary focus on resilience. However, a detailed cross sectoral and multidisciplinary conceptual framework, which links mitigation and adaptation strategies with health outcomes as a primary end point, has not yet been developed to guide research in this area. In this paper, we briefly summarize the burden of climate change on global public health, describe important mitigation and adaptation strategies, and present key health benefits by giving context specific examples from high, middle, and low-income settings. We then provide a conceptual framework to inform future global public health research and preparedness across sectors and disciplines and outline key stakeholders recommendations in promoting climate resilient systems and advancing health equity.

## 1. Introduction

Global climate change is a direct result of human activity on earth [1,2,3]. Human activities such as deforestation and combustion of fossil and biomass fuels have increased and continue to be the major contributor of greenhouse gas emissions (GHGE) such as CO_2_ and their accumulation in the atmosphere [1,2,4]. From 1990 to 2007, there was an increase of 45% in CO_2_ emissions from transport globally [5,6]. An additional 40% growth in CO_2_ emissions is expected by 2030 [5,6]. Additionally, global urbanization promotes consumerism [7] and the “throwaway culture [8]” of consumption and waste which is detrimental to the environment [9] and major source of pollutants. Urbanization is an integral part of economic development, but it occurs at the cost of environmental damages with increase in energy use and GHGE [10,11].

The continued increase in CO_2_ emissions poses a major threat to our planet due to climate-change-related extreme weather events (EWE) that negatively impact human health. Climate change increases the frequency and severity of EWE, such as heatwaves, wildfires, violent storms, floods, dust storms, and droughts; alters the pattern and distribution of vector-borne diseases; reduces crop yields; and contributes to population displacement from acute and chronic stressors such as resource scarcity and sea level rise [1,12,13,14], making it a highly significant to resultant and related diseases and injuries [15]. Multiple factors contribute to differential exposure and vulnerability to those negative health outcomes between individuals and communities, including socioeconomic factors, geographic residential area, and health status [16]. Individuals with increased risk of exposure and limited adaptative capacity are disproportionately affected, which adds to longstanding health inequities and disparities [17].

Fortunately, with collective action, the trajectory of current rising GHGE could be reduced [1,3,18]. However, the process requires coordinated international policy, supplemented by local policymaking and action to design and implement effective mitigation and adaptation strategies [18]. Some of the mitigation and adaptation strategies include multinational initiatives of the Paris Agreement from the United Nations Framework Convention on Climate Change (UNFCCC) 21st Conference of Parties (COP21) [19]. The Paris Agreement has the objective of keeping the global temperature below 2 °C or preferably below 1.5 °C above pre-industrial levels [20]. It requires all signatory Parties to submit their Nationally Determined Contributions (NDCs) to the UNFCCC explaining their specific plans in their efforts to locally support global climate change mitigation and adaptation efforts in the years ahead [21].

Globally, countries, states, municipalities, non-governmental organizations, and individuals have responded to the challenge of climate change with a range of mitigation and adaptation strategies. Climate change mitigation refers to “any actions or efforts taken to reduce or prevent the long-term risks of climate change on human life and property by reducing the sources or enhancing the sinks of GHGE” [22]. Major mitigation initiatives include reducing GHGE in sectors such as energy, transport, agriculture, water and sanitation; reducing methane (CH_4_) emissions through waste management and sewage treatment; embracing and promoting technologies that reduce or prevent anthropogenic GHGE; protecting and enhancing GHG sinks and reservoirs through sustainable management and conservation of forests, oceans, and wetlands; afforestation and reforestation; and rehabilitating drought affected areas [23]. In a nutshell, mitigation strategies intend to reduce GHGE and enhance GHG capture and storage, thereby reducing GHG concentration in the atmosphere and preventing additional adverse impacts of climate change on the environment and public health.

Climate change adaptation refers to “any activity that intends to reduce the vulnerability of human or natural systems to the impacts of climate change and climate-related risks, by maintaining or increasing adaptive capacity and resilience” [23]. This encompasses a wide range of activities from information and knowledge generation, to capacity development, planning and implementing proactive interventions to prevent or limit the impact of climate change consequences. Adaptation strategies can be found in a broad range of sectors with direct links to public health including water and sanitation, agriculture and food systems, forestry, fishing, flood prevention and control, and disaster prevention and preparedness, as well as public health preparedness [23]. Some specific examples include activities such as promoting water conservation in areas prone to increased water stresses; promoting crops resistant to heat and drought; promoting water saving irrigation methods in areas exposed to dry spells such as in the Sahara Desert area; implementing measures for flood prevention and management such as watershed management, wetland restoration and changing landcover to increase green infrastructure and reduce impervious surface particularly in areas prone to storm surge with increased risk of flooding; building more resilient systems with increased disaster prevention and preparedness measures such insurance schemes to cope with potential climatic disasters and creating early warning systems [23] and incorporating the World Health Organization (WHO) Guidance for Climate Resilient and Environmentally Sustainable Health Care Facilities into all health systems globally [24]. There are five adaptation stages: awareness, assessment, planning, implementation, monitoring and evaluation [25]. Overall, adaptation strategies seek to increase the resilience of systems to climate change and decrease the effects of future climate change events.

In summary, climate change mitigation strategies are essential to reducing global GHGE [26,27] while adaptation strategies are essential to alleviating the adverse effects of climate change events by increasing systems’ resilience to future climate change adverse effects [26,27]. While the primary goal of adaptation and mitigation strategies is to combat climate change and reduce its negative effects on the environment and societies, these strategies have also resulted in numerous health co-benefits or positive externalities. In this paper, we seek to catalogue and describe these positive externalities across world regions in order to consolidate existing knowledge and offer a conceptual framework to guide future work on mitigation and adaptation strategies and those positive externalities. We begin with a brief review of the climate change burden on global public health. We then describe major climate change mitigation and adaptation strategies with examples from different world regions, including high, middle, and low-income settings, in order to illustrate the global variation in strategies. We draw upon this body of work to present a conceptual framework to guide future research on public health benefits resulting from climate change mitigation and adaptation strategies around the world. We finally present major recommendations to stakeholders and give our concluding remarks.

## 2. Climate Change Burden to Public Health

### 2.1. Climate Change and Global Health

Climate change is a major public health concern with direct and indirect health harms that threaten exposed populations. The negative health effects are broad and include behavioral health disorders, diarrheal illnesses, heat-related illnesses, arboviral diseases, allergies, asthma and other respiratory disease exacerbations, cardiovascular diseases, and malnutrition [28,29]. Extreme heat may directly contribute to life-threatening heat-related illnesses and organ failure and wildfires to burn or inhalational injuries. Indirectly, inland, and coastal flooding may cause loss of access to health care services or supply chain disruptions. Millions of people are at risk from sea level rise alone [30]. The effects can be measured as excess morbidity and mortality [27] from non-communicable [31,32] or communicable diseases [33,34,35] but also as disability-adjusted life years, lost productivity and economic and health care costs as consequences of EWE. Limitations remain in accurately quantifying these diverse impacts as evidenced by a survey that demonstrated the number of excess deaths in Puerto Rico after Hurricane Maria was more than 70 times the official estimate [36].

The Intergovernmental Panel on Climate Change (IPCC), the leading scientific body on climate change, states that there will be “greater risk of injury, disease, and death due to more intense heat waves and fires (*very high confidence*), increased risk of undernutrition resulting from diminished food production in poor regions (*high confidence*), and consequences for health of lost work capacity and reduced labor productivity in vulnerable populations (*high confidence*) [37]”.

Other indirect relations to health are related to climate-fueled EWE, which include heat waves, droughts and desertification, severe storms, heavy precipitation, floods, tropical cyclones, wildfires, and dust storms [38,39,40,41]. For example, heat waves have been associated with all-cause mortality in France [42] with 15,000 excess deaths during an August 2003 heat wave. In the US, both extreme cold and extreme heat events have been associated with increased mortality [43] and risk of exposure is rising. In 2019, older adults in the US had 102 million more days of heat wave exposure compared with the prior 1986-2005 baseline [44]. In addition to extreme heat or cold, deserts and droughts from the rising temperatures [45] promote dust storms [46]. Dust storms have been associated with increased cardiopulmonary related hospital emergency visits [47], increased hospital stroke related admissions [48], increased intensive care unit admissions [49], increased cardiovascular and respiratory mortality in Spain [50], increased asthma related hospital admissions for children in Japan [51], and increased hospital admissions for pneumonia in Taiwan [52]. In addition to dust storms, flooding related changes in ecosystems modify vector transmission and risk of harm to humans [53,54]. Disruptions have caused important changes in the incidence, prevalence and distribution of infectious diseases [55], including vector-borne and zoonotic diseases, and water-and food-borne diseases [56], increased exposure and risk to *Vibrio vulnificus* [57] and increase in emergency department visits for children with diarrheal illness [58]. Drowning is a major cause of death, as seen after Hurricane Katrina [59]. Last but not least, wildfire smoke has been associated with increased self-reported symptoms, medication use, outpatient physician visits, emergency department visits, hospital admissions, and mortality for asthma and COPD [60,61], increased risk for congestive heart failure [62], and increased risk for overall mortality [63].

Kovats et al. (2005) summarized negative effects of climate change on human health by specific health outcomes such as increased illnesses and deaths related to heat-related illnesses during heat waves; increased mortality and morbidity related to air-pollution; vector-borne diseases such as malaria; water borne-diseases affecting disproportionately the poorer communities with poor water supply and sanitation; and food-borne diseases and food insecurity related health outcomes such as malnutrition due to draughts and decrease in food supply [64]. Additionally, climate change affects demand for health care and is linked to reduction in income [65], creating not only new health problems but also exacerbating issues in resource availability that contribute to conflict and migration [66,67].

Climate change effects are not homogenous in impact or distribution. While all people are at risk, specific populations are at increased risk of adverse health outcomes [41,44,68]. The effects of climate change frequently compound existing crises, such as a global pandemic and structural racism, and amplify social and environmental factors that influence health within institutions and communities, across nations, and on every continent [41,68]. Adverse consequences of climate change disproportionately affect communities with increased vulnerability and limited adaptative capacity such as those with low income (i.e., some communities of color, immigrant groups, and indigenous people), children and pregnant women, older adults, vulnerable occupational groups (i.e., outdoor workers), and those with underlying health conditions or disabilities [69] and exacerbate existing inequities and health disparities [70,71,72]. Even historical policies such as the 1910′s zoning and 1930’s redlining practices in the US have been implicated in the very structures that define vulnerability to resident exposure to heat [73] or greenspace [74] in urban environments and location of polluting oil and gas facilities in African American communities [75,76,77]. Thus, a public health approach represents an opportunity to advance health equity and intergenerational climate justice.

One specific example of differential health outcomes is respiratory infections that affect children of low-income and historically Black communities near polluting industrial facilities [78]. Exposure to air pollution from electricity generation is greatest for race/ethnicity even after adjusting for income in the US [79]. Other adverse impacts related to climate change on children’s physical and mental health include the following: asthma exacerbations and allergies; physical trauma from disasters; behavioral health disorders, including post-traumatic stress disorder (PTSD) after disasters and anxiety about the future; infectious diseases; and malnutrition and lack of clean water [80]. Even before birth, air pollution and heat exposure are associated with adverse pregnancy outcomes including preterm birth and low birth weight with the greatest risk being in minority populations and those with a history of asthma [81].

A nine-year-old girl’s death was recently attributed to air pollution in a landmark coroner case [82]. Air pollution related to vehicular traffic has been associated with decreased lung function in a community-based population prospective cohort study, after adjusting for tobacco smoke, asthma diagnosis, and socioeconomic status [83]. Similar findings were reported in Tokyo, Japan [84], and in another US community based cohort of 15 792 middle aged men and women where higher traffic density was significantly associated with lower forced expiratory volume after adjusting for potential confounders including demographic factors, personal and neighborhood level socioeconomic characteristics, cigarette smoking and background air pollution [85]. These effects have been associated with increases in chronic obstructive pulmonary disease (COPD) hospitalization and mortality in a population based cohort study (sample size, *n* = 467,994) in Canada [86], worsening of cardiovascular health outcomes in individuals with diabetes and coronary artery disease [87], and increased risk for total and cardiovascular mortality [88]. Most recently, air pollution was linked to COVID-19 mortality in the US [89]. Higher historical PM_2.5_ exposures were positively associated with higher county-level COVID-19 mortality rates after accounting for many area-level confounders [89].

The health care costs associated with climate-sensitive conditions is a growing concern for advancing health equity. An analysis of morbidity and mortality associated with just 10 climate-sensitive events that occurred in 11 US States during 2012 (i.e., wildfires, ozone air pollution, extreme heat, infectious disease outbreaks of tick-borne Lyme disease and mosquito-borne West Nile virus, extreme weather, impacts of Superstorm Sandy, allergenic oak pollen, and harmful algal blooms), estimated the total health-related costs from 917 deaths, 20,568 hospitalizations, and 17,857 emergency department visits at $10 billion in 2018 dollars [90]. The mortality costs ($8.4 billion) exceeded both the morbidity and lost wages costs combined ($1.6 billion) [90]. Another US analysis of health costs associated with six climate change–related events (i.e., ozone air pollution, heat waves, hurricanes, outbreaks of infectious disease, river flooding, and wildfires) between 2000 and 2009 estimated the health costs at more than $14 billion, with 95 percent due to premature deaths [91]. The estimated related health care costs was $740 million, reflecting more than 760,000 encounters with the healthcare system [91]. In 2020, there were a record-breaking 22-billion-dollar climate and weather-related disasters in the US which left at least 262 people dead [92]. Worldwide in 2019, nearly 98 million people were affected by disasters (97% climate and weather related) and at least 24,396 people died [93].

### 2.2. Climate Change and Global Injustice: Examples from Africa

While high-income countries tend to be most responsible for the proportion of global GHGE, low and middle-income countries suffer the most [71]. The continent of Africa is particularly vulnerable to the effects of climate change [37,94]. Sub-Saharan Africa is regarded as the fastest urbanizing region in the world with significant opportunity to ensure safety and protect lives and livelihoods [95]. The leading causes of morbidity and mortality in Africa, such as undernutrition, malaria, and diarrheal diseases, increase as temperatures and rainfalls become more variable [96]. In Mozambique, for example, studies have shown a link between increased temperatures and rainfall and previous malaria and diarrhea outbreaks [97]. Between 1997–2014, 7 million cases of diarrheal illness were attributable to reported temperatures rises in Mozambique [97]. In 2005, diarrheal disease was the fifth leading cause of death in Mozambique while malaria was responsible for 26% of all hospital deaths nation-wide [97].

Similarly, in Rwanda, a temporal and seasonal analysis of diarrheal illness incidence trends and climatic variations found an association between increase in diarrheal cases among children under age five and climate change dynamics from 2014–2018 [98]. An increase in temperature of one degree Celsius was associated with an additional 17 cases of diarrheal illness [98]. Additionally, in Rwanda, climate change related adverse events have been associated with increase in annual deaths, physical injuries, collapsing houses contributing to homelessness, and loss of crops exacerbating existing food insecurity challenges [99]. This also negatively affects mental health by increasing incidence of anxiety, stress and depression [99]. Moreover, consistent with previous studies, an analysis of daily health facility visits for enteric symptoms (diarrhea, gastroenteritis, or vomiting) and daily precipitation data for all under-five children in Rusizi, one of 30 districts of Rwanda, found a statistically significant association between extreme rainfall events and clinically diagnosed enteric infections [100]. Climate change exacerbates the movement of contaminants and water supply vulnerabilities and increases the risk of infections from extreme rainfall events [100]. In a similar fashion to Rwanda, the entire East African region has been affected by increases in vector borne diseases such as malaria that have been linked with increased temperatures together with changes in rainfall [101].

In addition to extreme rainfall events and rising temperatures in Rwanda and Mozambique, worsening air quality contributes significantly to the global burden of respiratory and cardiopulmonary diseases [38]. In South Africa, for example, the predominantly coal-fired power industry along with other industrial processes, domestic energy use and exhaust from vehicle emissions worsened the air quality [102,103]. An analysis of mortality burden attributable to urban outdoor air pollution in 2000 found that increased levels of air pollution caused 3.7% of the national mortality from cardiopulmonary disease and 5.1% of mortality attributable to cancers of respiratory tract in adults aged 30 years and older, and 1.1% of mortality from acute respiratory infections in children under five years of age [104]. Additionally, increased air pollution was associated with increased incidence of respiratory infections such as asthma and pneumonia [105].

Climate change is also a threat to global food security [106]. WHO has ranked malnutrition as the largest global health problem associated with climate change, particularly in low-income countries [107,108,109]. On the African continent, climate change increases the vulnerability of food production systems due to agricultural dependency on environmental conditions such as rainfall [110] and temperature, particularly in the sub-Saharan region [111,112,113]. In a 2010 literature review focused on climate change and food insecurity in Sub-Saharan Africa [114], climate change consistently predicted decreased crop productivity, land degradation, high market prices, negative impacts on livelihoods, and increased malnutrition. Those climate change induced fluctuations negatively affected food availability, food access, food utilization, and food stability [114]—the four pillars of food security [115].

Climate change exacerbates existing high rates of poverty and food insecurity in low-income countries such as Rwanda where 62% of its 12.5 million habitants live in extreme poverty on less than $1.25 per day [116]. Additionally, in 2018, over a third of Rwanda’s population experienced food insecurity and 35% of children under five years of age suffered from chronic malnutrition in 2018 [117], a slight decrease from 38% in 2015 [116], but still very high, and unacceptable. In Rwanda’s 2018 Comprehensive Food Security and Vulnerability Analysis (CFSVA), 40% of households reported experiencing weather related food security shocks such as drought, irregular rains, or prolonged dry spells which forced them in engaging in coping strategies such as harvesting immature crops and consuming seed stocks, exacerbating the vicious cycle of poverty [117].

Other country specific examples include Kenya, Uganda, and Tanzania, in East Africa and Nigeria, in West Africa. In Kenya, rising temperatures and declining rainfalls have been associated with childhood stunting since 1975 [18,112]. One study of 140 farmers in Uganda found that climate change events (e.g., flooding) were perceived as a major contributor to food insecurity by 95.5% of respondents [118]. In Tanzania, an analysis of climate change events (rainfall variability), food insecurity and human mobility in three villages located in the same district of Kilimanjaro found a positive correlation between rainfall shortage and out-migration and identified food insecurity as mediator in that relationship [119]. In Nigeria, a 2013 analysis of climate change events including temperature and rainfall from January 1971 to December 2009 in the Cross River State, prone to floods and oil spill hazards, found that climate change events were associated with rural household food insecurity, contributing to an estimated annual agricultural productivity loss of 67.7% [120].

Unmitigated climate change threatens the viability of organized human societies and represents a significant opportunity to protect health as a human right [1]. Right now, the international community has a time-sensitive imperative to reduce GHGE and prepare their own communities against the effects of climate change [121]. Average global temperature is expected to rise by 2 °C and 4 °C respectively by the years 2050 and 2100, as compared to the year 2000 baseline [18,122]. Yet, a different scenario is possible and waiting if we choose, whereby global warming would be stabilized between 1.5 °C and 2 °C and the worst of public health effects would be reduced or avoided [123]. On a global scale, governments and partners across disciplines need collective action to reduce GHGE; regionally, communities ought to adapt and build climate resilience based on local vulnerabilities and values rooted in public health and equity. A dedicated evidence-based research agenda with emphasis on mitigation and adaptation can drive programming, practices, and national and international policies that incorporate health. The range of these strategies is discussed below.

## 3. Climate Change Mitigation and Adaptation Strategies

### 3.1. Climate Change Mitigation Strategies

Climate change mitigation strategies include all actions to reduce GHGE such as limiting CO_2_ emissions by increasing usage of renewable energy, increasing the carbon sinks and reducing the use of fossil fuel energy [26]. Fossil fuel use can be reduced by enforcing energy efficiency measures such as building energy-efficient structures [26] with environmentally friendly and carbon-neutral materials including the use of earth blocks and earthen floors [124], which use little to no cement, one of the major contributors in global warming and climate change [125]. In the electricity generation as well as transportation sectors, the use of battery electric cars and electric heat pumps or gas burners, as opposed to oil burners can provide an economically viable venue to reduce the energy system’s reliance on carbon [126], therefore reducing GGHE. Carbon sinks can also be increased by planting trees to sequester carbon [87,88] particularly in urban areas.

Other global CO_2_ emissions reduction strategies include the implementation of multinational initiatives of the Paris Agreement [19] including countries’ specific Nationally Determined Contributions [127]. For example, China, the world’s largest CO_2_ emitter, intended to reduce its coal-fired power plants to less than 50% and 20% by 2020 and 2030 respectively [128]. Additionally, China increased the use of alternative renewable energy (solar, nuclear and wind power sources) and imposed strict regulations and penalties on companies that contribute to excessive air pollution [128]. China has embarked on a mission to achieve 20% non-fossil energy as a proportion of primary energy supply by 2030 [129]. A national work plan was put together to control GHGE, which includes investments in energy efficiency improvement to lower carbon emissions, increased investments in climate change research and development for both monitoring and forecasting, promotion of carbon emission trading, controlling emissions from the housing and transportation sectors, and increasing the national forest carbon sinks by adding an additional forest volume of 1.3 billion cubic meters by 2020 compared to the 2005 levels [130].

A recent study found that China is likely to achieve its emissions targets well in advance of 2030 and achieve its non-fossil target based on current policies [129]. This achievement assumes full and effective implementation of all current policies, including successful conclusion of power-sector reform, full implementation of a national emissions-trading system (ETS) for the energy sector and additional major industrial sectors after 2020 [129]. China’s mitigation efforts are expected to improve air quality, which can save 3000–40,000 lives annually in addition to the annual financial gain of over one billion RMB [131], equivalent to 140 million USD.

Other mitigation measures include carbon capture and storage, reducing non-CO_2_ gases and conservation and sustainable management of forests [127]. It has been estimated that forest and trees store a total of 643.2 million tons of carbon and sequester about 25.6 million tons of carbon per year in all 50 states in US [88]. In South America for example, the Brazilian government has a target of restoring and reforesting 12 million hectares of forests for multiple uses by 2030 to reduce carbon emissions [21]. A systematic policy approach was used in South Africa where the historic low energy prices attracted and supported energy intensive industries leading to high GHGE per capita [132]. In response to the devastating air pollution resulting from these GHGE, South Africa enacted laws and policies to mitigate climate change such as the Clean Air Quality Act in 2004 with components on air quality management, national standards for ambient air quality, listing activities and their respective minimum emissions standards, and GHGE reporting, among others [133]. Specific mitigation strategies include the use of clean coal technologies, nuclear power, power generation from waste incineration and the use of biofuels as well as the increased use of hydropower [132]. Other important actions include extra taxation on electricity produced from non-renewable sources and vehicles for burning fossil fuel [132].

### 3.2. Climate Change Adaptation Strategies

Climate change adaptation strategies include all activities that increase systems resilience to future climate change impacts such as rainwater harvesting, waste and sewage treatments, natural resources management, food security enhancement, social and human capital development and strengthening institutions [26]. Other initiatives include the promotion of reforestation and urban green spaces like public parks, community gardens, street trees and other urban green infrastructure solutions in cities such as adding quality urban landscape including sidewalks to manage rainwater [134] while improving neighborhoods walkability [135] and using green roofs and green walls in construction [136]. The provision of incentives for climate-resilient construction [127] such as encouraging the use of green roofs or walls would provide positive results. Green roofs and walls are important in cooling down city areas during the summer, capturing storm water, and increasing human well-being while enhancing biodiversity [137]. Green construction practices should also be incorporated into health care facilities, notably emergency units and acute care arenas, and other essential structures relied upon for operations during EWE. Ensuring existing facilities are retrofitted with this infrastructure and all new facilities are built with it will be important for policy making and development for employee and patient health and community resilience.

Adaptation activities cut across various economic sectors. The research sector, for instance, helps in strengthening the evidence-based decision-making process. China for example invested in research and development for early warning systems for extreme weather and in technologies for water saving as well as desalination of sea water [130]. South Africa has also established research on climate impacts in five sectors, namely biodiversity, agriculture, water, cities and health [133,138].

In the agriculture sector, various innovations are used to increase the resilience of agriculture systems and enhance food production value chains, food security and ultimately improve individual and community nutritional status. Examples of adaptative agricultural innovations include the introduction of management systems for erosion, drought enhanced irrigation and the introduction of new crops resilient to heat, drought, pests, and other various diseases [127]. In East Africa, drought resistant crops such as amaranth were introduced to benefit food security [139]. The promotion and improvement of amaranth production in East Africa has significant benefit potential for small-holder farmers in Africa, by providing a stable source of income and food for subsistence farmers while improving resilience to the climate change impact through the prospect of supporting the establishment of food and nutritional security [139]. An assessment conducted with South African farmers on agriculture adaptation to climate change reported that access to improved drought-tolerant seeds and efficient irrigation systems is the best way to cope with the changing climate [140]. In addition to crop resilience, changing food consumption habits would also reduce emissions from the agriculture sector and with co-benefits for health. Scholars in the United Kingdom have recommended that reducing livestock consumption and investing in other plant-based affordable, healthy and low carbon emitting diets would contribute to CO_2_ emission reduction in the agriculture sector [141]. Additionally, reducing livestock consumption translates into a reduction in intake of saturated fats, with known risk factors for chronic diseases such as obesity [141], cardiovascular diseases [142], type 2 diabetes and cancer [143].

A handful of adaptative strategies were implemented through partnerships with the United Nations Development Program and several African governments to reduce vulnerability to climate change [113]. Those strategies include agroforestry to improve the soil fertility and erosion management in Mali, Comoros, Mauritius, Rwanda, Guinea, Congo, Burkina Faso and Benin; soil and water conservation in Eritrea; and watershed rehabilitation and management in Ethiopia, Rwanda and Zimbabwe [113].

### 3.3. Dual Purpose Strategies (Mitigation and Adaptation)

Some strategies contribute to both adaptation and mitigation at the same time. Examples for instance are in the urban planning sector where urban greening (i.e., tree planting and parks creation) can help in mitigation through photosynthesis [144], while cooling cities during heatwaves and water absorption during flooding (adaptation) [145,146]. Another example is in the agriculture sector through climate-smart agriculture (CSA) practices [147,148]. CSA uses agricultural practices that sustainably enhances resilience and supports the achievement of national food security and development goals (adaptation) [149]. In addition to enhancing resilience by promoting better coordination between farmers, researchers, private sector, civil society and policymakers towards climate-resilient pathways [147], CSA enhances productivity, incomes and farmers’ resilience to climatic stresses, and reduces GHGE (mitigation) [150].

The CSA practices contribute indirectly to improving air quality and improving the health status of the population by easing the burden of air pollution related diseases such as respiratory and cardiopulmonary diseases [26]. Diversified farming and agroforestry are both examples of good CSA practices [151]. Agroforestry is used as a mitigation strategy in carbon sequestration but also as an adaptation strategy in increasing soil fertility and protecting the soil from erosion [152]. In West Africa for example, agroforestry parklands play an important role in buffering climate risks by sequestering carbon via photosynthesis [152]. Those agroforestry parklands also provide other positive externalities such as being used as medicine, food and recreational opportunities in green space [152]. Agroforestry also preserves and strengthens the environmental resource base of Africa’s rural landscapes [153] and enables the domestication of new tree crops to sequester carbon and increase air quality [154].

In addition to agroforestry in rural areas, urban and peri-urban agriculture and forestry (UPAF) has been recognized in both East and West Africa in mitigating climate change but also in alleviating poverty and enhancing food security in the long run [155]. UPAF has many other benefits such as improving air quality by using up the carbon, provision of biomass, which is the main source of cooking energy at home, and offsetting the urban heat island (UHI) effect through increased green space within urban areas and their surroundings, all enabling cities to become more resilient to adverse impacts of climate change [155]. Other UPAF benefits include storm surge protection, erosion control, flood regulation and microclimate moderation [155,156,157]. Those additional health benefits associated with UPAF remain largely under-investigated [155]. In addition to East and West Africa, community forests are used for both carbon sequestration and in biodiversity conservation in Nepal [158]. These efforts have other health benefits such as increased social capital and local communities’ livelihood opportunities, and enhanced access to food supplements such as roots, tubers, fruits and flowers [158].

The health sector is uniquely integrating into global climate change action by addressing the impact of its own GHGE (4.4% of net emissions [159]) and building climate-resilient healthcare systems [24,160]. Countries are committing to climate-smart health care and developing low-carbon technologies, risk mapping, waste reduction, early warning systems, strengthened infrastructure, actionable disaster preparedness plans, cross-sector collaboration, and health in policies that support the Sustainable Development Goals and Sendai Framework for Disaster Risk Reduction [161]. Researchers are highlighting the impacts of disasters on disease management to inform action plans and preparedness [162] and using implementation science for health adaptation for at-risk island nations [163]. The National Health Service has committed to net zero emissions [164]. Physicians are outlining agendas for the advancement of universal health coverage [165] and a pathway to net zero for emissions for all of health care [166].

Healthcare infrastructure and facilities are also being strengthened. The Maldives, a small island developing state, created a hospital vulnerability and assessment report [167], and Madagascar created a Climate Change and Health Diagnostic to address climate related health impacts and propose feasible solutions based on available resources [168]. Other countries have developed and implemented National Adaptation Plans that incorporate health [169]; Bangladesh is one example [170]. The Health and Climate Change Country Profiles detail specific threats and solutions [171]. Education on climate and health for health professionals [172] and policies that promote equity and equality, including gender equality [173], have engaged a diverse group of health professionals committed to systemic changes [174]. Health Care Without Harm (Global Green and Healthy Hospitals), Practice Greenhealth, and The Medical Society Consortium on Climate and Health are a few groups that are actively engaging students and practicing health professionals to advocate for environmental justice and health.

## 4. Positive Externalities of Mitigation and Adaptation: A Strategic Conceptual Framework

While climate change mitigation and adaptation strategies are essential in reducing GHGE (mitigation) [26,27] and alleviating the adverse effects of climate change by increasing systems’ resilience to future impacts (adaptation) [26,27], they have many other under explored health benefits [27,175]. Those strategies play both direct and indirect roles in preventing chronic diseases such as cancer, cardiometabolic diseases, and behavioral health disorders [27,175], preventing infectious diseases [176], increasing safety by reducing violence, anger, aggression, and crime [177,178], increasing food security [140] and generally supporting well-being. Of note, these health benefits are often seen in the shorter term and primarily enjoyed by the communities doing the interventions, while the benefits of climate change mitigation and adaptation strategies may take longer to be observed [179].

### 4.1. Methods

We drew upon the literature reviewed in the paper to develop a new cross sectoral and multidisciplinary conceptual framework. We identified associations and relationships between different variables connected through various pathways. If we take an example of agroforestry for instance, trees sequester carbon and reduce CO_2_ emissions. The process improves air quality which reduces risk for respiratory diseases and associated adverse health outcomes. All other variables and pathways are conceptualized in a similar fashion. We then created an excel dataset with key items from the literature reviewed throughout this paper. For the health outcomes, we chose relevant examples from the 2020 Global Burden of Disease review [15]. The dataset is attached in Appendix A. We then imported the data strings into RAWGraphs [180], a web application that created the alluvial chart presented in Figure 1. Figure 1 is a graphical illustration of the many different health benefits emanating from various climate change mitigation and adaptation strategies that are currently being implemented by different stakeholders in the literature.

### 4.2. The New Conceptual Framework

The conceptual framework is intended to represent the breadth of research work on this topic globally, emphasizing the complexity and directionality of relationships among strategies and outcomes. This conceptual framework offers an anchor and a guide for future studies in the growing and important area of climate change and public health research. Potential pathways are also proposed in the conceptual framework, drawing upon the work reviewed in this paper to follow each climate change (mitigation or adaptation) intervention (left column) through to its potential impacts on both climate change (middle column) and public health (right column). Columns provide pathways linking interventions with outcomes. This framework offers a conceptual view of the current state of knowledge in this area. It is meant to reflect the significant opportunity for growth in this space and has the potential to adapt as new discoveries and connections are made. Although we illustrate this framework through a discussion of key examples across selected sectors, the conceptual model can be used by all investigators with an interest in any of the interventions and how those interventions affect health outcomes through different pathways depending on their specific line of inquiry.

### 4.3. The Links between Public Health and Other Sectors in the New Conceptual Framework

#### 4.3.1. Urban Development and Green Infrastructure

A rapidly growing urban population puts the world in critical need for designing infrastructure to adequately respond to population growth [181]. Urban green infrastructure includes a wide range of natural elements in urban areas (sidewalks, bicycle or bike lanes, greenways, parks, gardens, green schoolyards, green roofs, woodlands, waterways, community farms, forests, and wilderness areas) [182]. Urban green infrastructure can serve as an effective strategy for climate change mitigation and adaptation [182]. Vegetation (in urban and rural areas) can capture and securely store carbon through biotic sequestration [158,182,183] while sidewalks, bike lanes and improved public transportation systems promote active means of transportation (e.g., walking and cycling) and reduce GHGE while promoting sustainable and resilient urbanization [184] and improving public health [185]. In general, proactive adaptation initiatives are also cost effective in the short and long term; the Department of Housing and Urban Development invested $930 million in a design-driven projects focused on infrastructure and disaster resilience following Superstorm Sandy [25].

Sustainable urban design with increased green infrastructure such as nature based solutions [186] reduce potential harmful exposures while offering other health benefits. Urban greenspace including public parks in urban and semi urban areas and community gardens help to filter the air by removing pollutants, reduce the urban heat island effect by cooling down towns and cities during warmer months and warming them up during the colder ones [187], increase water quality by reducing stormwater runoff [188], and consequently increase cities’ resilience to climate change impacts such as heat and flooding [145,189].

In addition to those anticipated climate change mitigation and adaptation benefits, there are other positive externalities for human health such as improved social well-being, physical and mental health [186,190,191,192,193,194]. Social wellbeing benefits include improved social connectivity [195], improved social relations, improved sense of place and increased social cohesion [18] and improved children’s socioemotional health [196]. Physical health benefits include improved self-perceived general health [197] and improved quality of life [198]. The improved green infrastructure also creates a conducive environment for increased physical activity (PA) behaviors [195,198], increased population fitness [199,200] therefore reducing sedentary lifestyles related diseases [201]. Urban greenspace has also been associated with reduced obesity risks [199,200], reduced risk for chronic diseases morbidity and mortality such as diabetes [200], cardiovascular diseases (CVD) and cancer [197,202,203], blood pressure and hypertension prevalence [199], and reduced CVD related mortality [204]. The mental health benefits include many positive mental and emotional health outcomes such as reduced stress [205], recovery from mental fatigue [190,206], and increased happiness [194].

Additionally, greenspace plays an important role in increasing safety by reducing levels of aggression and violence [177] and reducing crime [178] in inner city neighborhoods. In a natural experiment in Chicago, nearby greenspace was systematically related to reduced aggression against partners and children, measured by the validated Conflict Tactics Scale [177]. Additionally, an analysis of police crime reports in an inner-city neighborhood that examined the relationship between vegetation and crime found that greenspace proximity was associated with reduced crime reported for both violent and property crimes [178]. A systematic review of greenspace and crime outcomes found evidence of the impact of greenspace on a range of crime outcomes [207].

Finally, the novel coronavirus (SARS-Cov2) and the associated disease of COVID-19 have revealed new opportunities to leverage urban design strategies designed for climate change mitigation and adaptation to reduce infection risk while offering opportunities for outdoor physical activity and in person schooling. Outdoor greenspaces have been associated with a reduced risk of SARS-CoV2 infections and the resulting COVID-19 disease [208] by enabling physical distancing [209], a widely known and accepted measure for COVID-19 risk reduction [210]. Greenspace induced physical distancing mitigates the spread of COVID-19 by reducing the risk of transmission in non-crowded outdoor spaces compared to enclosed and crowded indoor spaces [211]. A review that investigated clusters of COVID-19 infections and their transmission settings linked very few infections to outdoor settings [212]. A more specific example of reduced risk in outdoor environment was in Oslo, Norway [213]. During Norway’s partial lockdown, outdoor environments facilitated physical distancing and reduced the risk of COVID-19 infection. Outdoor recreational activity increased by 291% relative to a three year average for roughly the same time period [213]. The increase in recreational use were greater in remote trails due to increased facilitation of physical distancing advantages with higher activity intensity for cyclists and pedestrians such as walking, running, hiking on trails with higher green views and tree canopy cover, indicative of the role of greenspace in increasing physical activity.

Additionally, urban greenspaces offer opportunities for nature exposure as well as outdoor and environmental education classrooms for students in urban neighborhoods. These outdoor classrooms reduce the risk of COVID-19 infection transmission by promoting physical distancing while enabling children’s social interactions and promoting their mental health while learning at the same time. Urban greenspaces offer additional opportunities for students recess in greener and healthier schoolyards which have been associated with children’s positive physical health outcomes including reduced sedentary behaviors [214,215], improved wellbeing and cognitive performance [216], reduced physiological stress [217], improved socioemotional health outcomes across numerous measures [196], and increased levels of physical activity [196,214,218,219]. Children’s physical activity is a well-established mechanism in preventing numerous adulthood diseases including chronic diseases such as cardiovascular disease [220,221,222], type 2 diabetes [221], overweight and obesity [220,222,223,224], and psychological disorders [225].

#### 4.3.2. Housing, Transportation and Agriculture

Health benefits are also prominent in the housing, transportation, and agriculture sectors. In the WHO’s Health in the Green Economy report series, health associated with the housing and transport sectors’ climate change mitigation and adaptation strategies were well described [189,226]. In the housing sector, those health benefits include reduced diseases such as asthma and COPD related to air pollution; reduced heat-related illnesses such as heat exhaustion and heatstroke; reduced extreme heat or cold exposure leading to illnesses such as hypothermia particularly in older adults and young children; prevention of vector and pest infestations; reduced home injuries; improved safe drinking-water and sanitation access; avoided use of toxic and hazardous construction materials; reduced vulnerability to floods, mud slides and natural disasters; support of slum redevelopment and physical activity friendly residential neighborhoods in fast-growing developing cities [226]. Investments in climate-friendly and energy-efficient housing can significantly reduce transmission of infectious diseases and aid in chronic disease prevention. Indeed, existing evidence suggests that that low-energy and climate friendly housing measures that encourage safe and energy-efficient home heating and appliances reduce exposure to mold and dampness and improve indoor air quality through better natural ventilation [226] and help prevent heart attacks, strokes, injuries, and other cardiopulmonary diseases.

In the transport sector, reduction of GHGE would be achieved through improved public transportation infrastructure and reducing the number of private vehicles on the roads and associated vehicular combustion [6]. These strategies have additional health benefits, which include reduction of diseases related to air pollution, reduced noise pollution and congestion [6], and reduced road traffic injuries [189]. They also foster resiliency to pandemics such as COVID-19 [227]. Improved public transportation infrastructure is also associated with increased physical activity and can be harnessed to reduce 3.2 million annual global deaths due to physical inactivity [189]. Public transportation also enhances health equity by improving mobility for women, children, older adults, people with disabilities and the poor who have less access to private vehicles, therefore improving their access to economic and social opportunities [189].

In the agriculture sector, CSA innovations result in soil conservation, increased crop production and other positive health outcomes [228]. In many African countries for example, climate change mitigation and adaptation efforts such as urban and peri urban agroforestry [155] results in poverty reduction and enhanced urban food diversity, therefore increasing food security and improved nutritional status. Other benefits from agroforestry parklands [152] include an increase in sense of place and cultural heritage, which have been linked with wellbeing, happiness and improved health outcomes [229]. Other innovations such as water saving irrigation address climate change [230], but also increases farm productivity [231] and therefore food availability. Additionally, in countries with limited access to water, water conservation can help ensuring water availability for basic sanitation purposes and hand hygiene, which are essential in preventing infectious diseases [232].

#### 4.3.3. Health Outcomes, Health Systems and Health Care Expenditures

Mitigation and adaptation strategies are in various stages depending on local stakeholders and resources. Health damages from US health care sector pollutants, for example, exceed 400,000 disability adjusted life years lost [233], which represents a significant opportunity for improvement. Climate-smart healthcare case studies highlight changes within healthcare facilities that positively affect health, such as saving more than 2000 tons of carbon emissions by serving vegetarian meals in Tzu Chi Hospital in Taiwan, reducing harmful anesthetic gases in Brazil, achieving energy independence in Gundersen Health in the US with local partnerships, and expanding renewable energy to rural health centers and clinics in Zimbabwe [161].

Historical examples in the United States have shown numerous positive health benefits associated with positive climate change policies. For example, a cost-benefit analysis of the US Clean Air Act from 1970 to 1990 found an association between the improved air quality and positive health outcomes including reduced incidence of cardiopulmonary diseases and other health benefits such as improvements in visibility and avoided damage to agricultural crops, implying improvement in food security [234]. In addition to those positive health benefits, other financial benefits ranged from $5.6 to $49.4 trillion against a cost of only $523 billion in 1990 dollars [234].

Researchers have projected that climate action would reduce noncommunicable [235] and communicable disease threats [236]. A recent study in Europe found that at least 51,000 premature deaths per year could be avoided by following guidelines on air pollution [237]. In a similar fashion to Europe, in the US, reductions in GHGE can bring health benefits of improved air quality and reduced premature mortality [238]. Public health benefits include avoiding 16,000 premature deaths for PM_2_._5_ related all-cause mortality per year and 8000 for ozone (O_3_) related respiratory mortality per year in 2050 [238]. Monetized benefits for avoided deaths from ozone and PM _2.5_ range between $45 and $137 per ton CO_2_ [238].

An assessment of three Latin American cities (México City, México; Santiago, Chile; São Paulo, Brazil) and one North American city, New York, showed significant positive health outcomes from reducing GHGE by using air pollution health impact factors appropriate to each city [239]. Health benefits included 64,000 avoided premature deaths, 65,000 avoided cases of chronic bronchitis, and 46 million person-days of avoided work loss or other restricted activity from years 2000 to 2020 [239]. In the US, one state (Wisconsin) reported that transition to 100% clean energy would have $21 billion per year in avoided health damages and create 162,000 net new jobs [240]. This would prevent 1910 early deaths, 34,400 asthma exacerbations, 650 heart attacks, and 650 emergency department visits for respiratory diseases.

Adaptation and resilience efforts highlight acute care services and chronic disease management during and after heat waves, tropical storms, wildfires, and other EWE. In India, a neonatal intensive care unit was relocated in the hospital to avoid persistent heat exposure; lower level locations were found to be protective for at-risk infants [241]. Within communities, healthcare systems have learned from prior events and are building resilience against extreme weather to minimize health and economic implications related to lost or disrupted access to health care services [242]. Lost revenue and jobs, delayed and cancelled surgeries, increased operating costs, supply chain delays and disruptions, evacuations, and harms to patients and staff in severe cases have been reported from climate-related EWE [243,244,245].

The Texas Medical Center convened following significant loss of research, study subjects, data, and hospital evacuations with Tropical Storm Allison [245] and outlined 11 lessons learned from that emergency evacuation process [246]. Stakeholders designed a multi-stakeholder community action plan and strengthened energy and external infrastructure to reduce damages for future events; when Hurricane Harvey came, hospitals remained open. Peebles Hospital in the British Virgin Islands is another example. Planners used the Pan American Health Organization Hospital Safety Index and Green Checklist for hospitals to strengthen infrastructure [247]. The hospital remained operational during and after Hurricane Irma hit the island as a Category 5 hurricane. The hospital even housed and protected displaced members of the community and disaster response operations. Minimizing loss of access to health care can be a key action to reduce sustained mortality as was seen for months after Hurricane Maria in Puerto Rico [36]. Health professionals and researchers are advocating for actionable disaster plans that include integrated chronic disease management plans as part of disaster plans, especially in vulnerable countries in the Caribbean [162].

## 5. Final Recommendations for Stakeholders

Authors propose a conceptual framework to guide present and future research efforts on climate change mitigation and adaptation and public health impact. The conceptual model presented in Figure 1 depicts critical pieces in thinking about specific interventions as they pertain to exposed populations and ultimately influence health outcomes. Researchers ought to challenge themselves to incorporate these key public health steps into their own scholarly efforts as they strive to optimize health for all. While the proposed framework is new, it builds off of existing guidance documents including checklists on health effects of mitigation actions have been proposed [68] and drive policy changes [248]. The framework also serves as a reminder to incorporate health benefits with policy rather than emphasis on economic assessment as has been described previously [238,249].

Having standards such as the Shared Socioeconomic Pathways (SSPs) Framework [250], or guiding documents such as the WHO Guidance for Climate Resilient and Environmentally Sustainable Health Care Facilities [24], can address health impacts via preparedness and response, including climate resilient health systems [251], and achievement of the United Nations Sustainable Development Goals. The ultimate goal of the SSPs Framework is to produce integrated scenarios that include socioeconomic and environmental conditions as affected by both climate change and climate policy, but health outcomes were not the primary intent of the framework [250]. Similarly, existing guiding document such as the WHO Guidance for Climate Resilient and Environmentally Sustainable Health Care Facilities is designed primarily for public health professionals or healthcare managers in hospital settings with a primary focus on resilience [24]. The new conceptual framework presented in this work builds off this critically important work through a systems thinking approach in order to guide public health research and mitigation and adaptation impacts in health-determining sectors, including agriculture, food systems and nutrition, water and sanitation, housing and urban development, and emergency management and disaster preparedness, energy and transportation, and health care delivery itself. The pathways identified can help researchers in developing their own conceptual and theoretical frameworks for specific questions aimed at looking at a particular mitigation or adaptation intervention impact on a specific health outcome of interest.

It is also important to recognize the positive work being done across all income settings, which can be potentially adapted and applied in other settings. Together, public health researchers will be instrumental in forming global partnerships and catalyzing the actions necessary to transform a planet sick with climate-related conditions to one of health and prosperity for generations to come through a salutogenic approach.

Final suggestions for stakeholders are as follows:Health should be incorporated into all policy creation and implementation with public health scientists and health care professionals engaged at each stage of policy development. “Health in all policies” or “health outcomes in all interventions” should be considered a gold standard moving forward.Government leaders should prioritize climate action within their cities, states, nations, and across borders focused on environmental justice and advancing the health of vulnerable populations.Climate education should be incorporated into schools and graduate studies to ensure a basic foundation of science across sectors and critical thinking skills are built. Application of the topic is encouraged for those interested in dual degrees in health professional studies.Research funding should incentivize climate smart initiatives, including green and climate resilient infrastructure, climate resilient agriculture and food systems, climate resilient transportation systems, climate resilient healthcare systems, and healthy environments in low, middle, and high-income settings with emphasis on developing tools, technologies, and models that accurately categorize risk and quantify health impacts from EWE.Rapid identification and implementation of solutions that reduce GHGE on a global scale and adapt and build resilient communities locally should guide those in positions of leadership and power.Collaboration across multiple sectors under this new framework should alleviate any duplication of efforts and ensure efficiency as we strive for an evidence-based and impact driven decision-making process to reduce health disparities and promote intergenerational equity.

## 6. Conclusions

There is commendable work on health impacts of climate change mitigation and adaptation strategies, but many of those benefits remain still un- or under-quantified in the literature [249]. This implies that the benefits reviewed in this paper are only a portion of many more benefits that still need to be investigated and applied across geographic locations. Building off of previously described work to integrate public health into climate change policy [252], an increase in inter-sectoral collaboration among scientists, health professionals, public health officials, and policy makers is warranted, and necessary, to create and sustain positive change. Conceptualization of benefits, in specific localized contexts at the national and subnational level, will continue to evolve our knowledge, while informing evidence-based decision and policy making processes with a research lens toward equity in different sectors such as agriculture, food systems and nutrition, forestry and biodiversity conservation, construction and infrastructure development, transportation and energy use, research development and innovation, housing, urban planning and development, and public health and healthcare systems, among others.

The conceptual framework presented in this paper is essential in future studies and investigations in climate change and public health that evolves as new information comes to light. We echo other scholars that integrating climate change adaptation into public health practice is essential [253]. Similarly, incorporating health outcomes into climate change intervention research will continue to advance our understanding of those interventions’ impacts on different health outcomes across different geographical and socio-economic contexts while supporting our planet’s resilience to climate change, and most importantly, the people on the planet. Public health investments in climate change interventions and vice versa, along with a rigorous quantification of the impact, will offer great benefits to the field and to planetary health.

## Figures and Tables

**Figure 1 ijerph-18-02481-f001:**
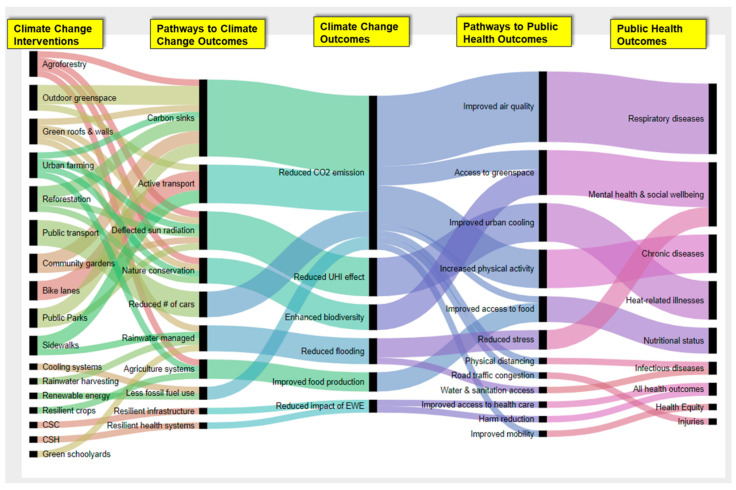
Conceptual framework to guide research on climate change mitigation and adaptation strategies and human health benefits. CSH: Climate Smart healthcare. CSC: Climate Smart Construction. EWE: Extreme weather events.

## Data Availability

No publicly available dataset was used in this paper, but we used information from literature to create a small dataset, used in the conceptual model. (See: Appendix A).

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
