# Peer review of "Positive Externalities of Climate Change Mitigation and Adaptation for Human Health: A Review and Conceptual Framework for Public Health Research"

_ijerph, 2021, doi:10.3390/ijerph18052481_

Round 1

Reviewer 1 Report

This is an important subject area.  

The paper provides a good synthesis of literature from a broad range of topics in the climate change and health field.  In addition, this is a very well reference paper.

The conceptual framework that is presented in the paper is useful and an important contribution.  Developing the model further and/or adding more depth to the recommended actions would strengthen the paper.

There are some statements made in the paper that would benefit from being more balanced.  

Author Response

Thank you for reviewing our manuscript! Your comments have greatly improved the current version of the manuscript. 

Reviewer 2 Report

In this paper, the authors argue that there are ancillary health benefits accrued from certain climate adaptation and mitigation initiatives and that these co-benefits should be actively considered. This is a robustly researched and cited paper and reflects a strong grounding in the literature.

The stated aim of the paper, to argue for greater understanding of how climate adaptation and mitigation efforts can be seen to also have human health benefits, is important. I am not sure if I agree that health benefits should be accepted to only be ancillary, however (although I understand the effort to interact with climate adaptation and mitigation literature) and wonder if the authors feel interested in making a stronger argument for an insistence that ‘health in all policies’ and in this case ‘health in all interventions’ should be considered a gold standard?

In the introductory paragraph, the authors first define what climate change is, what climate mitigation is and what climate adaptation is. The authors have identified that because climate change is largely caused by humans that human activity can effectively help to redress the undesirable impacts of climate change. The authors also offer clear and accessible descriptions of mitigation and adaptation. I feel, however, that these definitions are widely understood already and that this is a chance to deepen the argument. As a start, presenting adaptation and mitigation as arenas of activity without interplay sets up a false binary which can then foreclose analytical opportunity. Human causes and human solutions to climate change exist along a continuum, where human engagement reflects a continuum of engagement as well, from becoming aware to climate change communication to caring enough to taking action to investing in responses. This type of thinking also challenges the propensity to overly link adaptation and mitigation activities to technological fixes. As a result, I think that the authors should carefully consider how to offer a more nuanced introduction to climate change adaptation and mitigation as a way to set up their forthcoming argument around the benefits of actions and interventions for human health.

In Section 2. Climate change burden to global public health, the authors offer a comprehensive overview of the impacts of climate change to human health that have been well document and therefore sit robustly as observed effects in the literature. The authors, however, take a deficit-oriented approach to describing the impacts. In some cases, the authors identify sources of the health harms, for example, coal fired plants and focus primarily on the impacts but largely focus on impacts to populations or parts of the globe. I have a mixed reaction to this kind of presentation strategy and my concerns include the effect of developing in the readers imagination faceless communities of people who are passive recipients and victims of these impacts. There are also many health impacts that come from having to deal with these issues (e.g. mental health sequalae, DALYs etc). Further, when we don’t account for the strengths and assets of the people who are working with these situations, we then can also set up the locus of intervention as logically coming from the outside, often from technology provided by rich countries.

Section 3. Climate change mitigation and adaptation strategies, is divided into three subsections, one which talks about mitigation strategies, one that talks about adaptation strategies and one that talks about Dual purpose strategies (mitigation and adaptation). I am reading Section 3 through the title and stated intent of the paper Positive Externalities of Climate Change Mitigation and Adaptation for Human Health: A Review and Conceptual Framework for Public Health Research. Given the title, the work that I would expect Section 3 to present includes offering information about mitigation and adaptation activities in a way that explicitly identifies their benefits to human health and ideally shows the reader a pathway for how it benefits human health and to quantify these impacts. Perhaps the material could be presented through a gradient of strong, moderate and weak benefits to health or an identification of what interventions explicitly planned for, did not plan for or unexpectedly impacted climate change. Part of the reason why I am looking for an interpretation of benefit to human health is that I am trying to encourage the authors away from binary framings again, e.g. this set of interventions benefit health whereas these didn’t.  In addition, I think helping people see themselves as part of the solution and not part of the problem, e.g. identifying themselves along a contribution continuum has merit and certainly reflects some of the recommendations coming out of climate change communication literature.

Section 4. Positive externalities of Mitigation and Adaptation: A Strategic Conceptual Framework presents the framework itself. In my understanding of how to create a conceptual framework, it is important to not only conduct a robust literature review, as has clearly been done, but to also be explicit about the most important variables and relationships that are going to be highlighted within the framework, and why. For me, a methodology section would seem important to a paper like this. I would also like more information not only about how the framework was created but also how to use it. In terms of the material that is highlighted in this section, I would also like to know why the three topics were focused on in this section and if it is relevant to understanding or using the framework. I would also like to know more about how to use the model. For example, do I just take the variables presented in the framework and only look for them in my research space or is there a way to engage with the literature and other data in order to use the framework to identify priority areas in my research area etc. I would also like to know if there is a theoretical underpinning to the model or is it data driven? Or, how does one identify priority issues or populations, for example, if I am trying to tackle equity issues through my use of the framework? These are sincere questions as I really am keen to have a model to draw from.

Aesthetically and visually, I think that the framework is effective and, although it is slightly difficult to interpret, I applaud the authors for using a visual representational approach which illustrates how a range of individual activities actually integrate to create a large effect.

Section 5, Conclusion. I think that there are already many initiatives underway to integrate climate and public health and so this paper is not an essential starting point but rather is joining efforts underway to advance this work. I would say, however, that it is a contribution to directly speak to the areas of adaptation and mitigation and to invite a more active and intentional linking of these areas to public health, which really does require intersectoral efforts.  

This paper is well researched and has important material to offer to existing conversations on how responses to climate change are seen as having relevance to public health.  

Author Response

(The authors gave the same response as above.)

Reviewer 3 Report

Overall comment:

The authors do a good job in summarizing the burden of climate change on public health. They provide some clear solid evidence to this effect. Though there is some important emphasis missing. First, as I mention in my specific comments below the role of urbanization needs to be explored more to build a clear connection as to why the conceptual framework addresses the urban design and green infrastructure – why should mitigation and adaptation focus in this area? Second, the lack of discussion around equity in identifying and implementing these mitigation and adaptation interventions is quite apparent. In just a few places I note where this can be interjected:

Equity lens:

Lines 481-484: Reduced crime in green areas, but are these green areas already wealthy areas?

Lines 489-492: Similarly, increased green space and reduced covid-19 risk. Also crowding and multi-family households are associated with higher rates of covid-19, this is a socioeconomic and race/ethnicity issue as well.

Lines 554-547: Some mention of the income gap in relation to individual vehicles.

In the summarization of the burden of climate change on public health, there was some emphasis on areas that were more vulnerable than others (e.g. Africa) but in the discussion of the framework, there was no real identification of how mitigation and adaptation should be distributed based on protecting the most vulnerable – key tenants of public health (See 10 Essentials of Public Health).

Finally, there was no real discussion of what other frameworks are similar to the one presented. Some thoughts go to things like the Shared Socioeconomic Pathways (SSPs) which are used in climate change projections and account for mitigation. There may be some evidence in the latest U.S. National Climate Assessment and IPCC report on co-benefits to climate change mitigation and adaption. How do these discussions differ from the framework that is presented in this manuscript? Some clear identification of what is out there, versus what this framework adds or covers (if there are gaps) is needed to strengthen the paper.

Specific comments:

Lines 66-84: The authors may want to connect the dots full in their examples of climate change adaptation of how methods such as ‘water-saving irrigation’ have direct or indirect impacts on public health. The more focus on health will be in line with the focus of the manuscript.

Lines 128-165: Unsure why the focus is specifically on Africa here. While the authors identify Africa as particularly vulnerable, it is not the only vulnerable geographic region. It may benefit a reframing of what health vulnerabilities to climate change are in specific geographic locations given socioeconomics, geopolitical instability, or geographic/meteorological/topographic characteristics that increase health vulnerabilities in these locations. You could then follow up with some examples of Africa in particular – but initially, a broader identification of global health vulnerably is needed for context.

Lines 175-187: How are you linking vehicular traffic to climate change? Yes, transportation systems increase air pollution, but how is this related to changing climate – seems out of place in this section.

Section 3.1: I’m surprised that there is no discussion of the transportation sector in this section. Within the past few years, in the US at least, the transportation sector has overtaken energy as the largest contributor to GHG. Mitigation will not happen without changes in the transportation sector and this should be emphasized here.

Section 3.2: The examples in this section include terms like ‘initiatives’ ‘innovations’ ‘encouraging’ or ‘invested’. What specific evidence is there in the literature that these adaptive measures reduce negative public health impacts associated with climate change?

There are some examples provided, but they are secondary or tertiary – it would be beneficial to the reader to link specifically to how it reduces negative health outcomes. Epidemiological evidence would be a plus.

Section 4.1: There is some confusion to the reader with the focus on Urban Design and Green Infrastructure – the prior sections did not emphasize urban areas, although we know generally that urbanization is happening at an increasing rate. There was no full explanation previously in the manuscript about the potential connection to urbanization and impacts from climate change and how these areas may have increased negative health outcomes as a result. This should be added to better integrate the framework focus on this area.

Section 4.3: The examples the authors provide are centered on air quality and reduced greenhouse gas emissions. The authors then go on to say that “…impact assessments have rarely considered ancillary health benefits with climate change mitigation…”, which is not necessarily true. There has been a significant amount of work in assessing the co-benefits of reducing air pollution, it has just not been traditionally couched under ‘climate change mitigation’. The authors should acknowledge this and identify examples, non-air pollution (or non-GHG emission) mitigation, and more adaptation methods that emphasize interventions that have actually not considered health co-benefits if they wish to emphasize this point.

Author Response

(The authors gave the same response as above.)

Round 2

Reviewer 3 Report

The manuscript is much improved in both clarity and robustness of discussion around the framework.

Author Response

Thank you very much for your comments. We wanted to make sure that both robustness and clarity are top notch in our manuscript, to the best of our abilities and knowledge.